# Optical Differentiation of Brain Tumors Based on Raman Spectroscopy and Cluster Analysis Methods

**DOI:** 10.3390/ijms241914432

**Published:** 2023-09-22

**Authors:** Anuar Ospanov, Igor Romanishkin, Tatiana Savelieva, Alexandra Kosyrkova, Svetlana Shugai, Sergey Goryaynov, Galina Pavlova, Igor Pronin, Victor Loschenov

**Affiliations:** 1Institute for Physics and Engineering in Biomedicine, National Research Nuclear University MEPhI (Moscow Engineering Physics Institute), 115409 Moscow, Russia; ospanovanuar99@gmail.com (A.O.);; 2Prokhorov General Physics Institute of the Russian Academy of Sciences, 119991 Moscow, Russia; 3Department of Cryopreservation and Molecular Genetic Analysis, N.N. Burdenko National Medical Research Center of Neurosurgery, 125047 Moscow, Russiasergey255@yandex.ru (S.S.); pavlovagv@nsi.ru (G.P.);; 4Institute of Higher Nervous Activity and Neurophysiology of the Russian Academy of Sciences, 117485 Moscow, Russia

**Keywords:** Raman scattering, diffuse scattering, fluorescence, spectroscopy, intracranial tumors, cluster analysis

## Abstract

In the present study, various combinations of dimensionality reduction methods with data clustering methods for the analysis of biopsy samples of intracranial tumors were investigated. Fresh biopsies of intracranial tumors were studied in the Laboratory of Neurosurgical Anatomy and Preservation of Biological Materials of N.N. Burdenko Neurosurgery Medical Center no later than 4 h after surgery. The spectra of Protoporphyrin IX (Pp IX) fluorescence, diffuse reflectance (DR) and Raman scattering (RS) of biopsy samples were recorded. Diffuse reflectance studies were carried out using a white light source in the visible region. Raman scattering spectra were obtained using a 785 nm laser. Patients diagnosed with meningioma, glioblastoma, oligodendroglioma, and astrocytoma were studied. We used the cluster analysis method to detect natural clusters in the data sample presented in the feature space formed based on the spectrum analysis. For data analysis, four clustering algorithms with eight dimensionality reduction algorithms were considered.

## 1. Introduction

Recommendations for neuro-oncologic surgery include maximal tumor resection with minimal risk of functional complications with mandatory use of preoperative planning, including functional magnetic resonance imaging and tractography, microsurgical techniques, and intraoperative optics [1]. Reliable information on the volume of the resected tumor can be obtained using intraoperative imaging. The solution to this problem is mainly implemented through the use of optical systems (operating microscopes and endoscopes), intraoperative computed tomography, magnetic resonance imaging (MRI), intraoperative ultrasonography (iUSI), three-dimensional frameless ultrasound neuronavigation, neuronavigation systems, metabolic navigation, and various combinations of these methods. Since the possibilities of intraoperative MRI are limited and iUSI does not provide information on tumor metabolic parameters, the development and improvement of intraoperative optical neuroimaging and optical spectroscopy methods in neuro-oncology are relevant [2,3,4].

Currently, a multimodal strategy in the surgery of malignant brain tumors using electrical cortical stimulation, neuronavigation, and fluorescence is generally accepted [4,5,6,7]. The multimodal approach in intraoperative neuronavigation implies the combined application of modern possibilities of anatomical, physiological, and metabolic navigation, taking into account the individual characteristics of each patient and the equipment of the clinic in order to reduce the traumatic nature of surgical intervention on the brain. The application of metabolic navigation in neuro-oncologic interventions is possible in microsurgery, stereotactic biopsies and endoscopic interventions [8,9,10,11], using not only 5-aminolevulinic acid (5-ALA) but also other fluorophores such as chlorin-based fluorophores [12], fluorescein [13] and indocyanine green [14]. In the late 1990s, the first data on the possibility of using 5-ALA in neurosurgery appeared, which were mainly devoted to malignant gliomas [2,3,4]. Later, papers appeared on the possibility of simultaneous use of 5-ALA (per os) and fluorescein (intravenous administration) [15]. The issue of low-grade glioma fluorescence is controversial. In particular, a number of studies have shown a high variation of phytodiagnostics sensitivity in this category of patients, ranging from 0 to 40% [8]. In [16], in non-contrasting gliomas, the phenomenon of the glow of foci, which are zones of anaplasia, is noted during fluorescence according to the MRI data. At the same time, in “pure” Grade II tumors, according to some authors, there is no luminescence.

The most widely used method for intraoperative demarcation of tumors in neurosurgery today is the video analysis of Protoporphyrin IX (Pp IX) fluorescence in a special excitation mode with the violet light of an operating microscope. However, this method is only suitable for tumors with a high accumulation of fluorescent tumor markers (such as Pp IX). If the marker does not accumulate in the area of interest, which occurs in 40% of benign tumor surgeries [17], the surgeon must be provided with other means to distinguish tumor tissue from normal tissue. Of course, a tumor has a wide range of characteristic features (high proliferative status, hypoxia, high cell and nucleus density, destructuring of myelin) that distinguish it from normal tissue. An appropriate spectroscopic technique can be selected to determine each of these characteristics. However, tumor tissues at different tumor sites may not have all of these characteristics, and a more comprehensive intraoperative analysis should take them into account in order to achieve maximum efficiency in tumor tissue differentiation. Therefore, in this study, we use the previously developed combined spectroscopic technique as the basis for applying clustering methods [18]. Raman spectroscopy is one of the methods of vibrational spectroscopy, i.e., it allows one to obtain information about the structure of a molecule based on the vibrational sublevels of the molecule. This method is very convenient for biomedical applications because it does not require the introduction of any marker into the patient’s body and, therefore, does not depend on the peculiarities of marker accumulation, for example, which are already introduced for fluorescence analysis. Thus, we can obtain molecularly specific “signatures” of the tissues under virtually any condition. One disadvantage of this method is the weak Raman scattering cross-section compared to elastic scattering, which results in the need for long signal exposures. However, this is not a limitation for ex vivo analysis of freshly excised samples. For intraoperative analysis, there are a number of approaches to improve the sensitivity and speed of the method, which will allow further translation for the intraoperative approach.

Recently, several research groups have also focused their interest on the combination of different optical–spectral methods. For example, in the most recent study, a combination of histology based on stimulated Raman scattering and analysis of an autofluorescence signal with two-photon excitation was considered [19]. The authors obtained interesting results on the correspondence between autofluorescence signatures of tissues and their pathomorphological description; however, this study deals with the analysis of microscopic images of specially prepared samples. The main goal of our study is to develop a spectroscopic technique that could be used in real time in vivo. In this direction, a number of groups are also conducting interesting research; for example, in 2005, Santos et al. used Raman spectroscopy in the region of high wave numbers for fiber optic analysis of brain tissue [20]. Later, they conducted studies both in the fingerprint and high wavenumber regions of Raman spectra ex vivo on porcine brain tissue, demonstrating that essentially the same diagnostic information is obtained in the two wavenumber regions [21]. A major step towards intraoperative analysis was made in 2015 by Jermyn et al. using Raman spectroscopy in living human brain tissue [22]. They obtained spectra of tissues from glioma patients during neurosurgery, and the system successfully distinguished normal brain tissue from tumor tissue with 90% accuracy. In 2015 [23], in vivo Raman spectra of a normal brain, cancer and necrotic tissue were measured in 10 patients, demonstrating that real-time inelastic scattering measurements can distinguish necrosis from vital tissue (including tumor and normal brain tissue) with 87% accuracy, 84% sensitivity and 89% specificity. But here we are still considering the use of Raman spectroscopy alone, in isolation from the success of other optical–spectral approaches. In [24], the authors analyzed ex vivo autofluorescence and diffuse reflectance spectra of brain tissues and, using partial least square-linear discriminant analysis (PLS-LDA), showed differentiation of gray matter from tumor tissue with 89.3% sensitivity and 92.5% specificity. In the most recent work [25], the combined application of fluorescence, Raman and diffuse reflectance spectroscopies for the detection and classification of brain tumor and cortical dysplasia with a label-free modality was explored and showed discriminating tumor/dysplastic tissues against control tissue with 91%/86% sensitivity and 100%/100% specificity, respectively, while tumor from dysplastic tissue was discriminated with 89% sensitivity and 86% specificity.

The success of automatic classification methods in this application of optical spectroscopy leaves open the question of how exactly the optical–spectral signatures of tissues correspond to their morphological characteristics since all automatic classification methods are supervised methods; that is, they work on the basis of preliminary labeling of data based on the results of pathomorphological examination. Our study is devoted to the preliminary analysis of spectral data using clustering methods. These are methods for grouping unlabeled data in a given feature space on the basis of their similarity/proximity, according to some metric of the distance between them. The most commonly used metric is the Euclidean distance, but other metrics such as the Manhattan distance or the Minkowski distance can also be used. This procedure is interesting because it non-subjectively groups input observations (in our case, objects described by a vector of features computed from spectra) based on the natural similarity of their properties (spectral features). Therefore, in our study, we aimed to find naturally occurring clusters of some brain tumor tissues in the spectral feature space.

The authors of [26] emphasize that although cluster analysis is commonly used for data exploration, there are other possibilities for its use: class structure changes over time, too high cost of labeling samples, which makes it difficult to obtain large data sets, and gap filling in labeled data, which is often the case with spectroscopic chemical analysis data.

## 2. Results

### 2.1. Results of Dimensionality Reduction

The authors of [27] showed the importance of preprocessing techniques, as the resulting clusters varied in number, size and associated spectral features depending on the selected technique. They concluded that the spectral features seen in the Raman spectra could not be unambiguously assigned to tissue labels, regardless of the preprocessing method. Thus, the authors of this work urged transparency in the methodology and implementation of the preprocessing data. In our work, we develop a similar approach to analyze spectroscopic data from intracranial tumors, but we use not only Raman spectroscopy but also the data obtained from fluorescence and diffuse reflectance spectroscopy. The fact that we work with spectra of different natures forces us to perform some unification of the analyzed data at the first stage of processing. We cannot simply analyze the spectra point by point; we select characteristic regions from the spectra of all modalities, in which we calculate indices of the presence of certain components. These are included in our initial vector of features [28].

In this study, we propose to analyze the crucial differences in the appearance of the data in different feature spaces after dimensionality reduction by different techniques (Figure 1, Figure 2 and Figure 3). In Figure 1b, we can see which contributions of the original spectral components of the data vector entered the first and second principal components when the axes were rotated using the principal component method. The largest contributions to the first component were phenylalanine, proteins, hemoglobin, lipids, and cholesterol, as determined by Raman spectra. The second component mainly represented elastic light scattering, which, as shown in our previous studies, determines the decrease in the optical density of tumors during the degradation of tissues and subcellular structures in the process of tumor development. We also see a negative contribution of carotenoids to the second component, which are markers of healthy tissue.

If we consider which spectral features have the largest representation in the new axes after linear discriminant analysis (Figure 2b), we see that carotenoids and elastic light scattering also contribute to the first axis; i.e., it can be compared in physical meaning with the second axis of the principal component method. The methodological differences between LDA and PCA (LDA aims to maximize the separation between different classes, and PCA tends to find the direction of maximum variance of the whole data massive) explain why we see the same elastic light scattering in the second component, but with the opposite sign. This is one of the parameters that allow us to separate classes of tumors based on their optical density. We also see that the second axis is largely shaped by the positive contribution of phospholipids and the negative contribution of proteins.

For nonlinear methods of dimensionality reduction (Figure 3), the visualization of the contribution of the original spectral features to the new components is difficult, so we will further consider the physical meaning of the resulting partitions for these cases after applying clustering, according to the contribution of the original spectral components to the resulting clusters.

We also consider various options for combining dimensionality reduction techniques with cluster analysis. For each method of clustering, we have chosen the two best approaches of dimensionality reduction, taking into account (a) the clustering quality metrics (Silhouette and Davies–Bouldin indices) and (b) the distribution of samples with different diagnoses into separate clusters, so that there is at least one clear cluster with preferably one diagnosis.

### 2.2. k-Means Method of Cluster Analysis

The best k-means clustering in terms of combining the maximum of the Silhouette Index (SI) and the minimum of the Davies–Bouldin Index (DBI) (0.64 and 0.46, respectively) was obtained for the locally linear embedding dimensionality reduction technique (Figure 4a,b). However, in terms of distinguishing meningiomas as a separate class, this method does not appear to be optimal. In Figure 3, we see glial tumors indicated in the bar graph in orange, green, and red, and meningeal tumors indicated in blue. For the LLE technique with k-means clustering, there are no clusters preferentially populated by meningiomas (Figure 4b). The second cluster contains 50% meningiomas, which essentially represents the maximum sensitivity of their detection in this case.

For LDA with k-means, we can see (Figure 4c,d) that the third cluster is predominantly populated by meningeal tumors. Let us call it the meningioma cluster. Then, the sensitivity of meningioma detection will be 72%, and the specificity will be 82%. If we consider the first two clusters as glioma clusters, the sensitivity of the clustering to gliomas averages 82% for these two clusters.

Looking at the spectral characteristics inherent to the clusters on average, for the combination of k-means and LDA, we see the high level of carotenoids and oxyhemoglobin as the main differences for the 3rd cluster (represented mainly by meningiomas). Moreover, the first two clusters, populated by gliomas with varying degrees of malignancy, show differences both at the level of elastic light scattering (which is much higher in the first cluster) and in the content of lipids (higher in the first cluster) and hemoglobin (lower in the first cluster). Thus, based on its molecular characteristics, the first cluster is more likely to contain more benign glial tumors and areas from the perifocal zone of glioblastoma. To further investigate these differences, it is planned to conduct a more thorough analysis of the pathological picture of the corresponding samples.

### 2.3. Method of Agglomerative Hierarchical Clustering

From the point of view of clustering metrics, the best result was achieved when applying hierarchical clustering with LLE with the number of clusters equal to 3 (Figure 4a,b). However, as in the case of k-means, the sensitivity of this clustering method for meningioma did not exceed 50%.

On the other hand, agglomerative clustering with linear methods of dimensionality reduction yielded interesting results on the distribution of pathological diagnoses between clusters (Figure 5c,d). If we take the second cluster in the partitioning of the data after LDA as the meningioma cluster, the sensitivity of their detection is 70% and the specificity is 77%.

In this study, for the second cluster (meningiomas) obtained with clustering after LDA, we see a distribution of molecular features similar to Section 2.2, with high levels of carotenoids, proteins, cholesterol and oxyhemoglobin. Almost the only characteristic in which a cluster with gliomas is numerically superior to meningiomas is elastic light scattering for one of them (cluster No. 1). We also see an extremely low level of fluorescence in this cluster, which allows us to associate it with the most benign parts of gliomas. As mentioned above, the high heterogeneity of glioblastomas requires us to further analyze these tumors with this technique.

### 2.4. DBSCAN

For the DBSCAN method, the SI was maximum for the LLE and HLLE methods, but the DBI was also quite high for them. The SE method showed the best combination of indices.

In Figure 6, we can see the clusters obtained with this technique (Figure 6a) and the distribution of diagnoses in these clusters (Figure 6b).

However, this method failed to identify a cluster predominantly populated by meningiomas, forcing us to conclude that it is not suitable for our data distribution.

### 2.5. Fuzzy Cluster Analysis

Since in fuzzy clustering, we do not have an unambiguous correspondence between an object and a cluster but only probabilities of belonging, we assume that an object is in a cluster if its probability of being in it is maximal compared to its neighbors. It is also important to consider the probability of finding each type of tumor in each cluster. To calculate the probability of occurrence of each diagnosis in the cluster, the probabilities of occurrence of each object in that cluster were summed up, normalized to 100%, and then multiplied by the resulting conversion factor.

Let us consider P_ij_ the probability of an object i belonging to the cluster j. Then, for each cluster, the conversion factor is calculated as follows:K_j_ = 100%/∑(P_ij_).(1)

And then we recalculate the probability of belonging of each object to shares (fraction) for each diagnosis (diagnosis coefficient d) F_dj_, summing the shares of each object in the cluster for each diagnosis:F_dj_ = ∑(K_j_P_ij_).(2)

In terms of clustering objects with different diagnoses, the linear discriminant analysis method performed best. The third cluster in this partitioning is a good candidate to find meningioma (Figure 7). The sensitivity of their detection is then 63%. The specificity is 72%. For LLE and PCA dimensionality reduction techniques, the sensitivity of clustering with C-means to meningioma was below 50% (48% and 47%, respectively), which obviously cannot be used to differentiate meningioma from glial tumors.

## 3. Discussion

Among the clustering methods considered, methods such as k-means, hierarchical clustering and c-means provided a fairly uniform distribution of objects into clusters. The DBSCAN method considered our data as practically a single cluster, from which we can conclude that it is not applicable to our distribution of features.

The use of different dimensionality reduction methods has shown different results in terms of clustering characteristics such as silhouette index, etc., and in terms of the distribution of different diagnoses into different clusters (Table 1 and Table 2). In the former case, the LLE and HLLE methods almost always showed the best results. However, they produced clusters that were dense, but almost evenly populated by each diagnosis. In the second case, the palm of superiority among the dimensionality reduction methods was taken by LDA, which in two cases (for clustering methods such as k-means and hierarchical clustering) led to obtaining a sensitivity for meningiomas above 70%, and 63% for fuzzy clustering.

Since cluster analysis works on unlabeled data, the separation of glial and meningeal tumors into separate clusters indicates their natural differences in the considered feature space formed as a result of the identification of characteristic bands in Raman, fluorescence, and diffuse reflectance spectra. The obtained results can be used both for the recovery of missing data and for the stage of data preprocessing when using classification algorithms on labeled data.

Our results are in good agreement with the results of [27], despite the fact that they analyzed only Raman spectra without additional modalities and without pre-filtering the data according to biochemical components. In addition, in our study, we present a comparative analysis of various combinations of dimensionality reduction and clustering methods, which allows us to draw conclusions about the nature of the data distribution and the advantages of some approaches over others. The studies described in [9] are instrumentally closer to our work. The authors also combine several spectroscopic modalities. However, in their study, only one method for differentiating normal tissue and dysplasia on labeled data was considered, whereas, in this study, we solved the problem of finding spectral differences between tumors of different origins. Further continuation of our work will be the development of a decision support system based on the automatic classification of multimodal spectral data. The results obtained in this study will be used to select the optimal method for dimensionality reduction and missing data recovery.

## 4. Materials and Methods

A He-Ne laser (Biospec, Moscow, Russia) was used to excite a fluorescence signal from biopsy samples, which was observed using LESA-01-BIOSPEC fiber-optic spectrometers (Biospec, Russia) with optical edge filters to attenuate the excitation radiation (632.8 nm) at the spectrometer input to the level of the fluorescence signal. A halogen lamp was used as a white light source to obtain a diffuse reflectance signal. The spectra were recorded using Uno software version 1.0.7 (Biospec, Moscow, Russia) installed on a laptop computer. The exposure time was 100 ms for the fluorescence signal, and 10 ms for the diffuse reflectance signal.

Raman scattering was excited with a 785 nm Ramulaser-785 laser (StellarNet, FL, USA) and observed with a Raman-HR-TEC-785 fiber-optic Raman spectrometer (StellarNet, FL, USA) with 4 cm^−1^ resolution. The laser radiation and the Raman scattered light were delivered using a confocal fiber-optic probe. The laser power was 150 mW measured at the output of the laser source. The Raman spectra of each sample were recorded in a series of 10 measurements with 30 s exposure for each measurement. The background signal was measured prior to each series. Both the background and the Raman spectra were measured in a darkened room [18].

The studies were conducted in the Laboratory of Neurosurgical Anatomy and Preservation of Biological Materials of N.N. Burdenko Neurosurgery Medical Center (Moscow, Russia) on tumor tissue samples removed during neurosurgery and placed in saline immediately after removal. The minimum sample size was 3–4 mm. After the spectral measurements, the samples in formalin were sent for pathomorphologic examination. Samples from 54 patients with diagnoses of glioblastoma (n = 28), meningioma (n = 12), astrocytoma (n = 9), and oligodendroglioma (n = 5) were examined. Between 1 and 4 biopsy specimens (93 tissue samples in total) were taken from each patient, followed by verification by pathomorphologic examination. Raman spectra, fluorescence of 5-ALA-induced protoporphyrin IX, and diffuse reflectance spectra in the 500–600 nm region were measured for all samples.

Optical–spectral measurements of each biological sample were performed in two steps including (Figure 8):Registration of diffuse reflectance spectra in white light and fluorescence spectra with 632.8 nm laser as an excitation source using the LESA-01-BIOSPEC spectrometer;Registration of the spontaneous Raman spectra of a sample excited at 785 nm using the Raman-HR-TEC-785 spectrometer.

Feature filtering removes features (wavelengths, wave numbers, and peak positions in relation to spectroscopy) that may contain noise or non-valuable information. This procedure allows us to reduce the dimensionality of the data and focus on the valuable information. The authors of [29] found that the choice of features is fundamentally important for the outcome of the classification, and they were unable to achieve class separation when analyzing the full spectra. In the current work, we used the approach to feature filtering that we proposed earlier [28]. There, it was shown that the step of pre-filtering features before applying feature projection methods to reduce dimensionality significantly improves the classification results. Some physical values corresponding to the used features are presented in Figure 9.

Dimensionality reduction techniques can be divided into linear and non-linear. Linear methods include principal component analysis (PCA) and linear discriminant analysis (LDA). Non-linear methods include multidimensional scaling (MDS), isometric mapping of objects (Isomap), local linear embedding (LLE), Hessian locally linear embedding (HLLE), spectral embedding (Laplacian Eigenmaps, SE), and t-distributed stochastic neighbor embedding (t-SNE).

PCA is used to project a higher-dimensional data matrix onto a low-component subspace. It reduces the set of variables to a smaller set of orthogonal, and thus independent, principal components in the direction of maximum variation, i.e., it reduces the dimensionality and preserves the most significant information for further analysis.

The main goal of linear discriminant analysis (also called Fisher’s linear discriminant) is to find “axes of discrimination” that optimally classify data into two or more classes. LDA is closely related to PCA (Principal Component Analysis) in that both look for latent axes that compactly explain the variance in the data.

The main difference between PCA and LDA is that LDA is a supervised method and PCA is an unsupervised method. PCA looks for predictions that maximize the variance, and LDA looks for predictions that maximize the ratio of interclass variance to intraclass variance (Figure 10). The data can be projected into a new dimensional space using the axes found by LDA. In the new multidimensional space, each observation will have fewer variables (lower dimensionality), and at the same time, observations belonging to the same class will form clusters, and each cluster will be clearly distinguished from the others [30].

Multidimensional scaling (MDS) seeks a low-dimensional representation of the data in which the distances closely match the distances in the original multidimensional space.

Isometric feature mapping (Isomap) projects the data into a lower dimension while maintaining geodesic distance (rather than Euclidean distance as in MDS). The geodesic distance is the shortest distance between two points on a curve.

Locally linear embedding (LLE) looks for a lower-dimensional projection of the data that preserves distances within local neighborhoods. It can be thought of as a series of local principal component analyses that are compared globally to find the best non-linear solution.

Hessian local linear embedding projects the data into a lower dimension, preserving local neighborhoods like LLE, but uses the Hessian operator to better achieve this result.

Spectral embedding uses spectral methods to reduce dimensionality by matching adjacent inputs and adjacent outputs. This preserves locality, but not local linearity.

t-SNE converts the affinities of data points into probabilities. Similarities in the original space are represented by Gaussian joint probabilities, and affinities in the nested space are represented by Student’s t-distributions. This allows t-SNE to be particularly sensitive to local structure.

Clustering is the process of grouping a set of physical or abstract objects into classes of similar objects. We can say that clustering is successful if the data in one cluster is very similar in one (or more) parameter, and has little similarity between data from other clusters.

Cluster analysis methods can be divided into two main categories:hard clustering: each object belongs to only one of the clusters;fuzzy clustering: each object belongs to each cluster to some extent; i.e., the algorithms fix the probabilities of a given object belonging to each cluster, and these probabilities must sum to “1”.

Also, we can consider another approach to classify them based on the mechanism of grouping data:centroid-based clustering;hierarchical clustering;density-based clustering;distribution-based clustering.

The k-means algorithm is the simplest and one of the most widely used cluster analysis methods. In addition to the data, this method requires additional input parameters, such as the number of clusters (K) and the initial center of the cluster.

In this work, we also used agglomerative hierarchical clustering. There are two main types of hierarchical clustering algorithms: agglomerative (bottom-up) and divisive (top-down) approaches. Divisive algorithms work according to the top-down principle: at the beginning, all objects are placed in one cluster, which is then divided into smaller and smaller clusters. More common are bottom-up algorithms, which start by placing each object in a separate cluster, and then the clusters are merged into increasingly larger clusters until all sample objects are contained in a single cluster.

DBSCAN (density-based spatial clustering of applications with noise) is a widely used algorithm for distinguishing closely spaced but fairly dense clusters of complex shapes.

When using fuzzy clustering methods, each observation eventually belongs to more than one cluster with a given probability. One of the widely used soft clustering algorithms is the fuzzy c-means clustering (FCM) algorithm, which gives better results for overlapping data sets compared to k-means clustering.

For hard clustering algorithms, we used three metrics of clustering quality: Silhouette Index (SI), Calinski–Harabasz Index (CH), and Davies–Bouldin Index (DBI).

The Silhouette Index is the average silhouette coefficient for all clusters, calculated using the average distance within a cluster and the average distance to the nearest cluster. This score ranges from −1 to 1, where the higher the score, the more well-defined and distinct the clusters are.

The Calinski–Harabasz Index is calculated using inter-cluster variance and intra-cluster variance to measure differences between groups. As with the Silhouette Index, the higher the score, the more clearly defined the clusters are. This score is boundaryless, meaning that there is no “acceptable” or “good” value. However, it can be used to compare different clustering methods.

The Davies–Bouldin Index is the average similarity of each cluster to its most similar cluster. Unlike the previous two metrics, this score measures the similarity of clusters, meaning that the lower the score, the better the separation between clusters.

For soft clustering evaluation, on the other hand, we used its own metrics based on the probability distribution of data in clusters: Partition Coefficient (PC) and Partition Entropy Coefficient (PEC). The PC measures the degree of homogeneity within each cluster, whereas the PEC measures the degree of heterogeneity between clusters. It is defined as the negative sum of the product of the fraction of data points in each cluster and the logarithm of the fraction of data points in each cluster.

Both PC and PEC range from 0 to 1, with higher values indicating better clustering results. A PC value close to 1 indicates that the data points are well clustered, whereas a PEC value close to 0 indicates that the clusters are very heterogeneous.

Since we observe a noticeable separation in the space of reduced dimensionality of objects defined by the pathomorphologist as meningiomas against the different glial tumors, it is reasonable to introduce parameters demonstrating how well meningiomas are distinguished into a separate cluster using different methods. For this purpose, in each partition, we call the clusters in which meningiomas predominate over glial tumors meningioma clusters. And in this case, we calculate the sensitivity of the clustering method for meningiomas (the number of meningiomas falling into the “meningioma cluster”) and the specificity (the number of non-meningiomas falling into the “non-meningioma clusters’’).

Obviously, there is no need to make this analysis meningioma-centric; we can consider the results in terms of the identifiability of glial tumors. Since they are fairly evenly distributed in clusters, it makes sense to consider them in general. The sensitivity of meningioma detection corresponds to the specificity of glioma detection, and vice versa.

## 5. Conclusions

In this study, cluster analysis methods were applied for statistical processing of data obtained from biopsy specimens of intracranial tumors. The data were obtained using Raman, fluorescence and diffuse reflectance spectroscopy. An optimal combination of dimensionality reduction and cluster analysis techniques was sought to detect natural differences between samples in the spectral feature space. Both linear (PCA and LDA) and non-linear approaches (MDS, LLE, HLLE, Isomap, spectral embedding, and t-SNE) were investigated. Cluster analysis was also performed using several methods, including hard clustering methods such as k-means, agglomerative hierarchical clustering and DBSCAN, and fuzzy c-means clustering technique.

After considering all possible combinations of the mentioned methods, we selected the two best dimensionality reduction algorithms for each clustering algorithm by considering two approaches to assessing the quality of clustering. The first approach was based on the clustering quality metrics (Silhouette and Davies–Bouldin Indices) and the second approach used the information about the distribution of samples with different diagnoses into separate clusters so that there would be at least one clear cluster with preferably one diagnosis.

From the point of view of the first approach, the best results were obtained using the agglomerative hierarchical clustering algorithm with the local linear nesting algorithm as the dimensionality reduction method. However, the sensitivity of this clustering to different diagnoses was low. The second approach to assessing the clustering quality gives us the optimal combination of LDA with k-means (72% sensitivity to meningeal tumors and 82% sensitivity to glial tumors) and the second-best combination of LDA with agglomerative clustering (70% sensitivity to meningeal tumors and 78% sensitivity to glial tumors).

The results of the study of the clustering algorithms of the obtained data allow us to conclude that they can be naturally divided into subsets of meningeal and glial tumors. For further differentiation of glial tumors, we plan to continue the study using classification methods on labeled data.

## Figures and Tables

**Figure 1 ijms-24-14432-f001:**
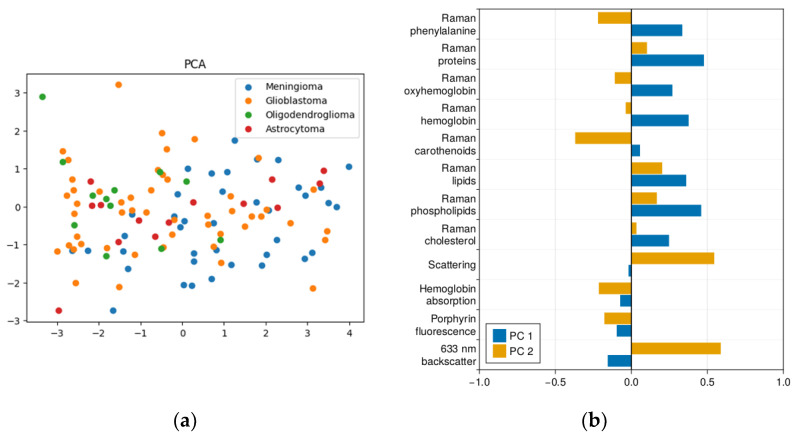
(**a**) Principal component analysis (PCA); (**b**) loading of principal components.

**Figure 2 ijms-24-14432-f002:**
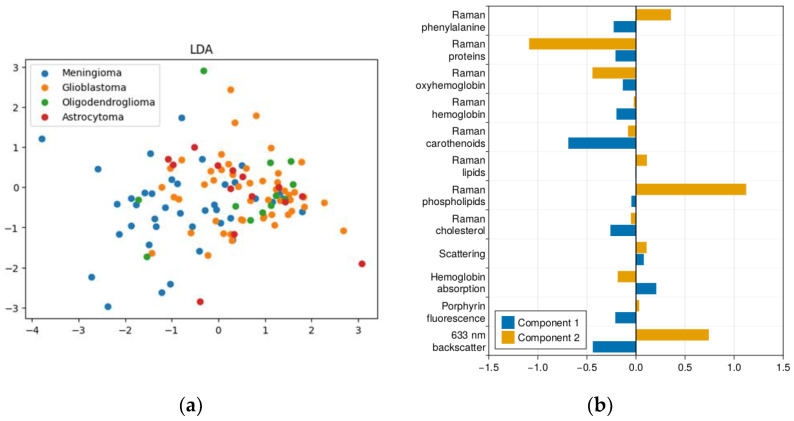
(**a**) Linear discriminant analysis (LDA); (**b**) LDA scalings.

**Figure 3 ijms-24-14432-f003:**
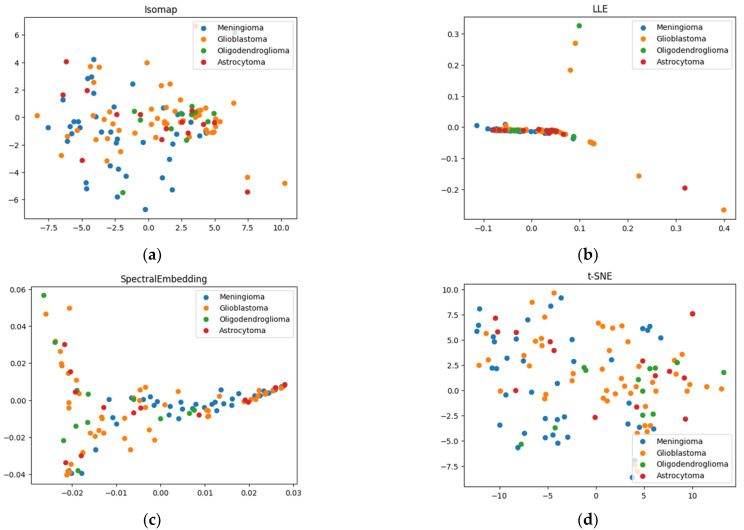
Non-linear approaches to dimensionality reduction of prefiltered spectroscopic data: (**a**) isometric feature mapping (Isomap); (**b**) locally linear embedding (LLE); (**c**) spectral embedding (SE); (**d**) t-distributed stochastic neighbor embedding.

**Figure 4 ijms-24-14432-f004:**
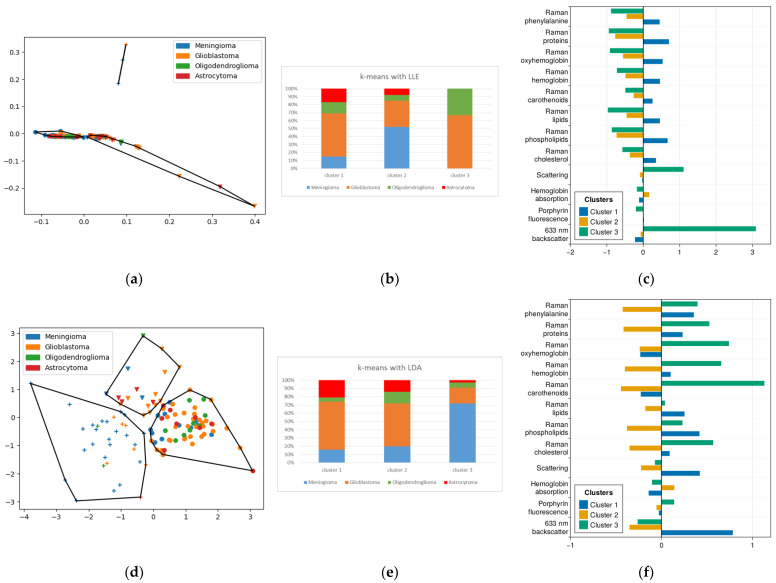
(**a**) K-means clustering results for LLE-reduced data, points with the same shape belong to the same cluster outlined by polygon; (**b**) histogram of diagnosis distribution for (**a**); (**d**) k-means clustering results for LDA-reduced data; (**e**) histogram of diagnosis distribution for (**c**); (**c**,**f**) mean normalized values of features corresponding to each cluster.

**Figure 5 ijms-24-14432-f005:**
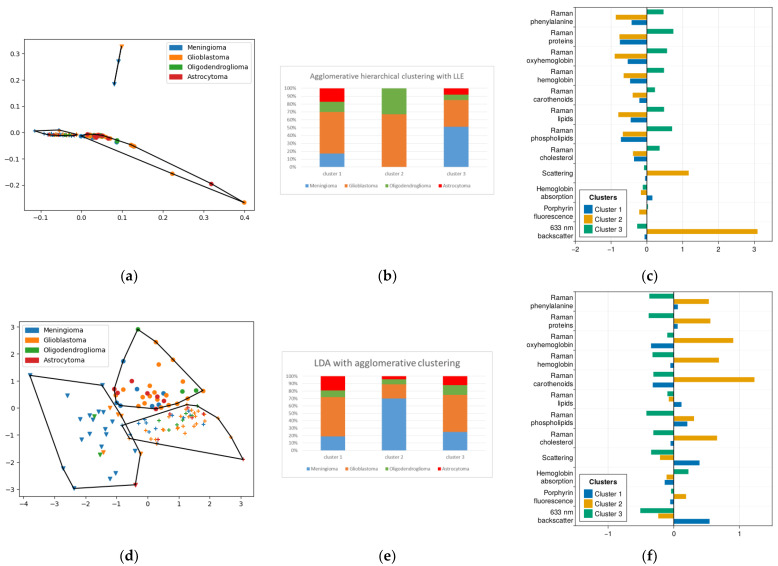
(**a**) Agglomerative clustering results for LLE-reduced data, points with the same shape belong to the same cluster outlined by polygon; (**b**) histogram of diagnosis distribution for (**a**); (**d**) agglomerative clustering results for LDA-reduced data, points with the same shape belong to the same cluster outlined by polygon; (**e**) histogram of diagnosis distribution for (**d**); (**c**,**f**) mean normalized values of features corresponding to each cluster.

**Figure 6 ijms-24-14432-f006:**
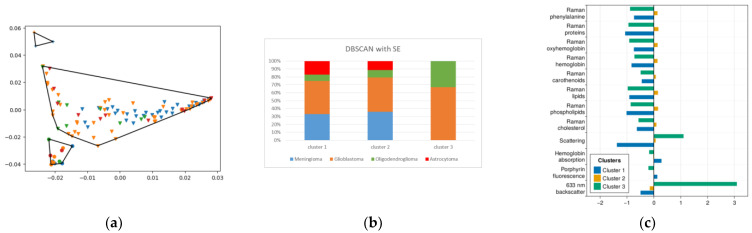
(**a**) DBSCAN clustering results for SE-reduced data, points with the same shape belong to the same cluster outlined by polygon; (**b**) histogram of diagnosis distribution for (**a**); (**c**) mean normalized values of features corresponding to each cluster.

**Figure 7 ijms-24-14432-f007:**
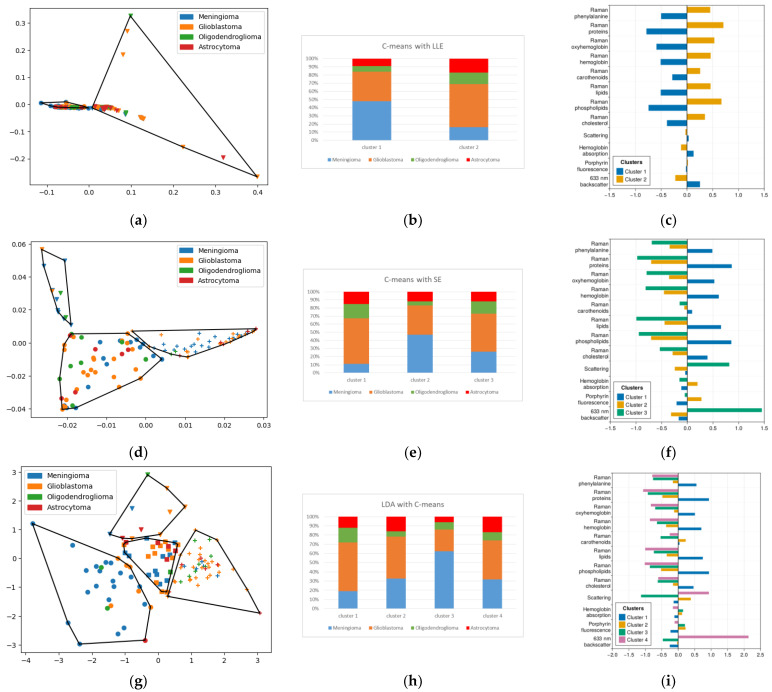
(**a**) LLE with C-means (55 objects in 1st cluster; 60 objects in 2nd cluster), points with the same shape belong to the same cluster outlined by polygon; (**b**) histogram of LLE with C-means; (**d**) SE with C-means (52 objects in 1st cluster; 12 objects in 2nd cluster; 51 objects in 3rd cluster); (**e**) histogram of SE with C-means; (**g**) LDA with C-means (48 objects in 1st cluster; 28 objects in 2nd cluster; 29 objects in 3rd cluster; 10 objects in 4th cluster); (**h**) histogram of LDA with C-means; (**c**,**f**,**i**) mean normalized values of features corresponding to each cluster.

**Figure 8 ijms-24-14432-f008:**
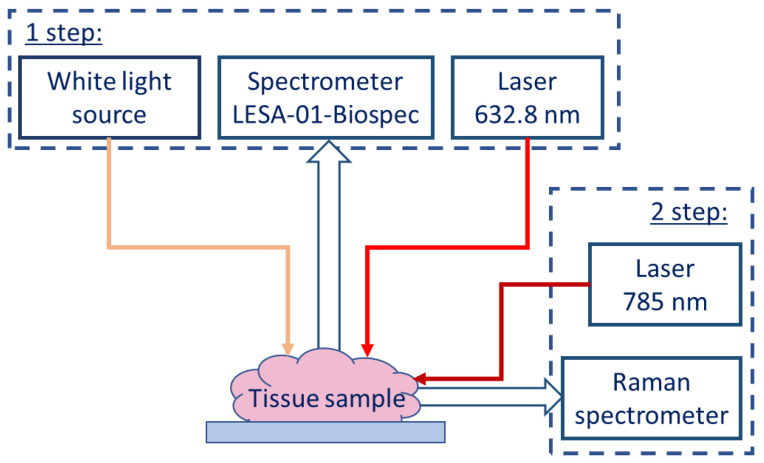
Scheme of spectroscopic analysis of fresh biological samples of intracranial tumors.

**Figure 9 ijms-24-14432-f009:**
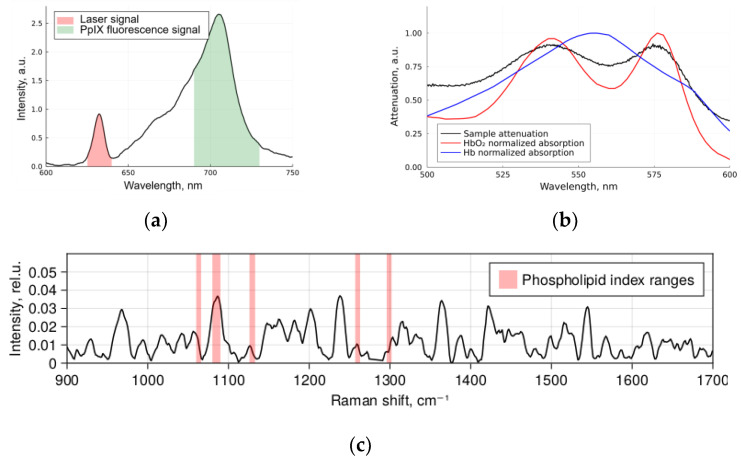
Spectral parameters used as features: (**a**) backscattered 633 nm laser and Protoporphyrin IX fluorescence; (**b**) hemoglobin absorption; (**c**) processed Raman spectrum with highlighted ranges corresponding to phospholipid Raman signal.

**Figure 10 ijms-24-14432-f010:**
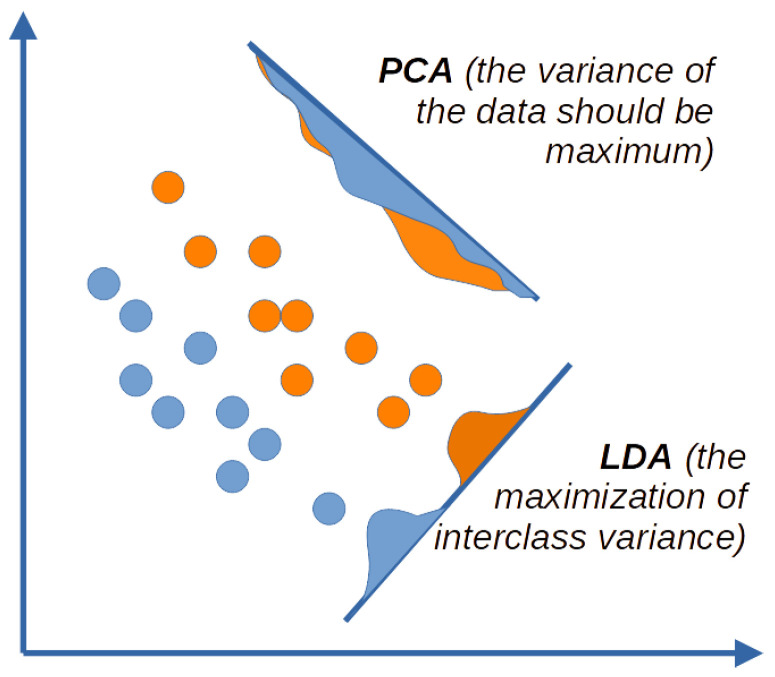
Difference between PCA and LDA approaches to dimensionality reduction. Points in different colors belong to different classes.

**Table 1 ijms-24-14432-t001:** Metrics of quality for hard clustering algorithms.

Algorithm	Silhouette Index	Calinski- Harbasz Index	Davies- Bouldin Index	Sensitivity/Specificity of the Meningioma Differentiation
LLE with k-means	0.64	256.68	0.46	52%/92.5%
LDA with k-means	0.37	125.57	0.89	72%/82%
LLE with agglomerative clustering	0.75	148.17	0.39	51%/91.5%
LDA with agglomerative clustering	0.32	101.58	0.99	70%/78%
Spectral embedding with DBSCAN	0.40	76.52	0.56	36.1%/83.5%

**Table 2 ijms-24-14432-t002:** Metrics of quality for fuzzy-c-means clustering algorithm.

Algorithm	Partition Coefficient	Partition Entropy Coefficient	Sensitivity/Specificity of the Meningioma Differentiation
LDA with C-means	0.11	0.26	63%/72%
LLE with C-means	0.16	0.12	48%/84%
Spectral embedding with C-means	0.14	0.18	47%/81.5%

## Data Availability

The raw data supporting the conclusions of this article will be made available by the authors, without undue reservation.

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
