# Peer review of "Optical Differentiation of Brain Tumors Based on Raman Spectroscopy and Cluster Analysis Methods"

_ijms, 2023, doi:10.3390/ijms241914432_

Round 1
Reviewer 1 Report
The given text discusses a scientific research project focused on the analysis of biopsy samples of intracranial tumors using various combinations of dimensionality reduction and data clustering methods.
The text lacks context about the significance of the findings in the broader field of intracranial tumor research or medical diagnostics. More details and references should be added.
The authors should decide "Davis or Davies" appears spelled differently. Please check the whole article.
How were the two best dimensionality reduction algorithms selected for each clustering algorithm? The text doesn't prove a clear connection between the current research and any existing literature or studies in the field.
Author Response
Thank you very much for your attention to our work.
All your comments were extremely useful to us and we hope that this has allowed us to improve the quality of our work.
The medical context analysis was added to introduction section with 16 additional references. We have also corrected the spelling of the Davies surname. Regarding the selection of optimal dimensionality reduction methods, our approach is described the end of the section 2.1: “For each cluster analysis, we have chosen the two best approaches to reduce the dimensionality, taking into account (a) the clustering quality metrics (Silhouette and Davies-Bouldin indices) and (b) distribution of samples with different diagnoses into separate clusters, so that there is at least one clear cluster with preferably one diagnosis”.
Best regards,
Tatiana Savelieva
Reviewer 2 Report
The introduction can be improved by comparing the Raman spectroscopy with other methods.
Figures must be improved; it is hard to understand. As we know Dimensionality reduction refers to the process of reducing the number of attributes in a dataset while keeping as much of the variation in the original dataset as possible, Section 2.1 should be improved with an explanation of the physical meaning of the figures for the shake of the general audience.
Authors should also provide the background calculations in the supplementary information. The physical meaning of all the analyses 2.2, 2.3, 2.4, and 2.2 are missing, and how these results are helpful should be provided. due to increasing interest in Raman spectroscopy in various diagnostics.
Minor editing is required
Author Response
Thank you very much for your attention to our work.
All your comments were extremely useful to us and we hope that this has allowed us to improve the quality of our work.
We expanded the description of the Raman spectroscopy method in the introduction, and also significantly changed the data presentation in section 2.1, presenting and analyzing the loading diagrams of new components for linear methods of dimensionality reduction. In the following subsections of the results section, we supplemented the visualization of clustering with diagrams of the average values of spectral features for each of the resulting clusters.
Best regards,
Tatiana Savelieva
Round 2
Reviewer 2 Report
The authors have employed the comments in the revised manuscript. I would recommend minor editorial checks before accepting the manuscript for publication.
Authors should be careful before final submission, and publication.
Author Response
We double-checked the manuscript and made grammar improvements.